# Network Biology-Inspired Machine Learning Features Predict Cancer Gene Targets and Reveal Target Coordinating Mechanisms

**DOI:** 10.3390/ph16050752

**Published:** 2023-05-16

**Authors:** Taylor M. Weiskittel, Andrew Cao, Kevin Meng-Lin, Zachary Lehmann, Benjamin Feng, Cristina Correia, Cheng Zhang, Philip Wisniewski, Shizhen Zhu, Choong Yong Ung, Hu Li

**Affiliations:** 1Department of Molecular Pharmacology and Experimental Therapeutics, Mayo Clinic College of Medicine and Science, Rochester, MN 55905, USA; weiskittel.taylor@mayo.edu (T.M.W.);; 2Mayo Clinic Alix School of Medicine, Mayo Clinic College of Medicine and Science, Rochester, MN 55905, USA; 3Department of Computer Science, Duke University, Durham, NC 27708, USA; 4Department of Chemistry, Biochemistry and Physics, South Dakota State University, Brookings, SD 57006, USA; 5Department of Molecular Cell and Developmental Biology, University of California, Los Angeles, CA 90095, USA; 6Department of Biochemistry and Molecular Biology, Mayo Clinic College of Medicine and Science, Rochester, MN 55905, USA

**Keywords:** gene dependency, systems biology, systems pharmacology

## Abstract

Anticipating and understanding cancers’ need for specific gene activities is key for novel therapeutic development. Here we utilized DepMap, a cancer gene dependency screen, to demonstrate that machine learning combined with network biology can produce robust algorithms that both predict what genes a cancer is dependent on and what network features coordinate such gene dependencies. Using network topology and biological annotations, we constructed four groups of novel engineered machine learning features that produced high accuracies when predicting binary gene dependencies. We found that in all examined cancer types, F1 scores were greater than 0.90, and model accuracy remained robust under multiple hyperparameter tests. We then deconstructed these models to identify tumor type-specific coordinators of gene dependency and identified that in certain cancers, such as thyroid and kidney, tumors’ dependencies are highly predicted by gene connectivity. In contrast, other histologies relied on pathway-based features such as lung, where gene dependencies were highly predictive by associations with cell death pathway genes. In sum, we show that biologically informed network features can be a valuable and robust addition to predictive pharmacology models while simultaneously providing mechanistic insights.

## 1. Introduction

Despite therapeutic advances, many diseases lack actionable targets, and a reliable strategy to identify new therapeutic agents is urgently needed. In translational research, gene dependencies are targeted to cause cell death in proliferative diseases including pathogenic infections and cancers. Gene dependencies are defined as genes that a cell needs for growth or continued viability and can be measured by growth restriction or cell death when the gene is inhibited [1]. The advent of targeted therapeutics dramatically increased the demand for novel specific gene targets that can alter a disease trajectory since numerous strategies for identifying these high-value targets have emerged, along with new concepts that help encapsulate and identify promising drug development targets. These approaches aim to identify genes with a high dependency in pathogenic cells so that pharmacologic inhibition is maximally effective. In an effort to increase knowledge of oncologic gene dependencies, the Broad Institute and the Wellcome Sanger Institute conducted and aggregated high throughput in vitro dependency screens on a wide array of cancer cell lines into a Cancer Dependency Map (DepMap) [1]. Others have contributed to this effort and explored alternative screening strategies, determinants of specific dependencies, and relationships between individual gene dependencies. Since its inception, DepMap has spurred the discovery of new key oncology targets and biological mechanisms [2,3,4,5].

Because of the significant resources needed to create such dependency maps and the immense heterogeneity of cancer, numerous groups have attempted to create predictive in silico models of gene dependency for high throughput target prediction in new tumor samples. Existing approaches have focused on inputting genomics and transcriptomics into deep neural networks or random forests for dependency prediction [6,7]. Other groups have focused on interpretability through visible neural networks and identifying unique mechanistic features such as expression dosage-based dependencies [8,9].

In parallel with the increased understanding of gene dependencies, network biology has emerged as a powerful tool for interpretability and mechanistic discovery from large datasets. Biological networks have enabled the discovery of numerous pharmacological interventions, particularly in oncology, because they allow for aggregative systems behavior to be captured in actionable formats, which has also translated to new therapeutics [10,11,12,13,14]. Based on the success of machine learning and network biology, we combined the two to predict and understand the regulation of gene dependencies. Rather than ignoring prior biological knowledge or using it to constrain model design as done in previous studies, we used biological network measurements as predictive features that embed high dimensional external biological data and then understood their importance through deconstructing predictive algorithms. In this way, we identified high-value predictive features that point to novel mechanisms of gene dependency coordination in specific cancer histologies. 

## 2. Results

### 2.1. Network-Based Features Are Predictive of Binary Gene Dependency

DepMap is an expansive dataset that measured 19,183 gene dependencies via CRISPR screening across numerous cancer cell lines [1]. To understand if network features can contribute to gene dependency prediction, we constructed network feature-driven artificial intelligence (AI) models. For each cancer type, we used the well-established network construction algorithm ARACNE to create regulatory networks for each cancer lineage in DepMap using Cancer Cell Line Encyclopedia RNAseq data (Figure 1) [15,16]. ARACNE constructs networks by detecting pairs of genes whose expression values are statistically dependent on one another. We calculated four categories of network parameters for each gene in each lineage-specific regulatory network: (1) traditional network features, (2) the number of cancer hallmark neighbors, (3) the sum of weights, and (4) the length of the path to cancer-associated genes Figure 1). Category 1 features included the degree of connectivity, closeness centrality, betweenness centrality, the sum of adjacent edge weights, and the average neighbor degree. This group of features was termed the traditional network features. Category 2 captured the cancer activity local to each gene by calculating the number of each of the 10 cancer hallmark genes present in the immediate neighborhood of each examined gene [17]. Finally, using the set of cancer-associated genes from the Cancer Gene Census, we computed the length of the shortest path from each gene to each cancer-associated gene as well as the sum of edge weights along this path for category 3 and 4 features, respectively [18]. With each of these feature groups established and measured for all possible cancer types and gene targets, we then examined the ability of each feature to predict whether a lineage depends on a given target gene (Figure 1). 

Lineage dependency was predicted with an array of methods: AdaBoost, Decision Trees, Gaussian Process, K Nearest Neighbors, Linear Support Vector Classifier (LVC), Logistic Regression, Random Forest, Ridge, Stochastic Gradient Descent optimization with LVC, and Radial Basis Function Support Vector Classifiers. Each was tested using our network features, gene expression, and both groups combined [19]. Accuracy was measured via the F1 score $\left(\frac{2\times precision\times recall}{precision+recall}\right)$ which encapsulates both precision $\left(\frac{N_{TP}}{N_{TP}+N_{FP}}\right)$ and recall $\left(\frac{N_{TP}}{N_{TP}+T_{FN}}\right)$. Because of this, F1 score reports model success in a more balanced way since only a minority of genes in each cancer lineage are deemed dependent. A gene was said to be dependent if its dependency score was less than −1 on average for the given cancer histology per Dempster et al. [7,20,21]. In most machine learning architectures, network features and expression had comparable performance, but for some, expression outperformed network features by a few percentage points (Figure 2A). In many cases, network features demonstrated higher variability on cross-validation than expression, which indicates that network-based prediction performance varied across different genes. Adding expression and network features together did not provide additional predictive performance in any model, but all models, regardless of input features or model type, performed with F1 scores greater than 0.90, indicating that network features have as much predictive utility as expression. Further, from the high accuracies across model architectures, we concluded that network features are as flexible as model inputs as expressions. Importantly, model accuracy was consistent across cancer types (Appendix A). Logistic regression performed well for all cancers, with lung cancer having a slightly lower performance (Figure 2B). Given the high interpretability, minimal computational load, and robust accuracy of logistic regression, we used logistic regression in our subsequent expansive testing of network biology features. 

### 2.2. Network-Based Prediction Was Stable across a Range of Network Construction and Dependency Hyperparameters

Dependency scores are presented by DepMap on a continuous scale, and initially, we used the DepMap-defined cutoff of −1 to call each average cancer dependency score dependent or non-dependent. DepMap scores were generated such that nonessential reference genes had a median score of zero. To ensure model robustness, we examined the effect of varying DepMap cutoffs on the F1 score. DepMap thresholds ranging from −1.50 to 0.25 were tested. F1 scores were poor (<0.5) until the cutoff decreased to under −0.25. Below −0.50, the curve plateaued, and only small changes to the F1 score were appreciated with further decreases in the DepMap cutoff (Figure 3A). F1 score significantly decreased as the cutoff increased to >0.05, indicating a greater difficulty in predicting close to neutral dependencies or gain of function knockouts (Figure 3A). The cancer specificity of this finding was tested, and again we found that performance was cancer-agnostic (Appendix A). From this, we concluded that network prediction is not sensitive to dependency cutoffs < −0.25 and thus moved forward with the previously established dependency cutoff of −1 [7,20,21]. 

We then pivoted to examine the effect of network pruning cutoffs on predictive performance. ARACNE constructs networks that are fully connected with the mutual information (MI) score between genes as edge weights [2]. Network pruning is often used on large networks to distill the network to the strongest interactions. Here we used a MI score of 1.5 for our initial testing and varied pruning cutoffs (0–25), which dictate the size of output networks. Predictive performance (F1) was not significantly affected by varying pruning from zero to 25. Performance was highest and most consistent with pruning cutoffs of less than five (Figure 3B). The lack of performance drop at smaller cutoffs showed that network features combined with logistic regression were robust to irrelevant edges in under-pruned networks (Figure 3B). The performance drop after five was likely due to the elimination of pertinent network interactions at this cutoff. 

### 2.3. Traditional Network and Biological Hybrid Features Encoded Overlapping Information Predictive of Gene Dependency

In order to understand the importance of each feature group, new logistic regression models were constructed that removed a single feature category at a time and generated ablation models. The F1 performance was compared across these single ablation models after five-fold cross-validation. We noticed that despite losing an entire feature class, all models had similar performance (Figure 3C). Given this result, we proceeded to create models with pairwise ablations (Figure 3D). Similarly, even the pairwise ablation models retained a similar performance to the full model. From this, we infer that despite being conceptually varied, dependency information was redundantly encoded in the feature groups allowing for robust prediction despite feature class ablation. This redundancy indicates that the underlying predictive mechanisms of gene dependencies manifest themselves both through general topology and relationships to cancer genes/hallmarks in regulatory networks.

The ablation of feature classes had a similar effect on most cancer types, but for lung and hematopoietic histologies, the ablation experiments changed the accuracies by a few (3–5%) percentage points (Appendix A). In both cancer types, the removal of the classical network features was the least disruptive to F1 accuracy, but these changes were not statistically significant. This indicates that traditional network features possibly encoded less gene dependency information than the other feature classes. The ablation models were also more variable in many cases than the full model, which hints at different genes relying on different classes of features as within each cross-validation fold, different genes are used to test the model. Similar results were seen with the double ablation models, which showed the same changes for lung and hematopoietic cell lines (Appendix A). Both categories within the CCLE contain diverse specific histologies (ex., lung: mesothelioma, non-small cell lung cancer, and squamous cell cancer), which could be contributing to their accuracy variability because the models are not able to capture such divergent biologies within the same ARACNE network.

### 2.4. Specific Features within the Larger Feature Classes Demonstrate Histology Specific Importance

To delve deeper into the important features within our models and examine potential biological implications, we created ablation models that remove single features and noted the change in model performance in the setting of this feature’s absence. For traditional network features, we noted that centrality measures and the degree of neighbors failed to provide additional performance to any histology’s predictions (Figure 4A). In contrast, features that focused on connectivity, such as the sum of adjacent edges and degree of connectivity, contributed to several tumors’ predictions (Figure 4A). Interestingly, we noticed the co-occurrence of high feature importance for both of these metrics in specific histologies (Figure 4A). The theory of centrality-lethality states that central genes in networks tend to be essential to cellular function, but in our results, these simple degree centrality measures outperformed more advanced centrality measures, which infer a node’s “centrality” in a network by quantifying relationships beyond first-degree neighbors [22].

Ablation models for each cancer hallmark revealed unique histology-specific results. The ablation of resisting cell death, activating invasion and metastasis, sustaining proliferative signaling, and evading growth suppression affected the performance of lung cancers’ dependency predictions uniquely (Figure 4B). Architype analysis on small cell lung cancer intra-tumor heterogeneity found that specific subclones within small cell lung cancer (SCLC) tumors optimize their phenotype preferentially towards proliferation, and this polarization leads to specific therapeutic vulnerabilities [23]. Here we see that these vulnerabilities are likely determined by their proximity to the identified cancer hallmarks, and genes within these hallmarks may be coordinating gene dependencies in lung cancers. Skin cancer was also affected by the ablation of activating invasion and metastasis, which was the only significant predictive pathway (Figure 4B). Skin cancer develops as it progressively invades the dermis and epidermis, and here we observe that these same pathways are crucial for coordinating gene dependency and cell survival [24].

Shortest and least-weighted paths also showed tumor-specific patterns of feature importance (Appendix A). Tumor lines from the autonomic ganglia showed several features that had higher importance compared to the other tumor types. The shortest paths to CANT1, CREB1, NIN, and TLX1 were the most important features in autonomic ganglia cancers (Figure 5A). Least-weighted path features were also important in autonomic ganglia tumors, with CREB1, FHIT, and BRD4 being the highest among several important least-weighted paths (Figure 5B). In a systematic analysis, autonomic tumors had the second highest number of CREB1-regulated genes upregulated out of over 20 tumor types [25]. CREB1 has not been specifically interrogated as a drug target in these cancers, but our analysis corroborates existing research that it plays a crucial regulatory role in autonomic tumors by recovering it from two separate feature sets [25]. In soft tissue tumors, the shortest paths to HEY1 and FOXO3 had the highest importance (Figure 5A). HEY1 fusions are often found as oncogenic mutations in chondrosarcoma, but their role in modulating gene dependency has not been specifically interrogated [26]. FOXO1 has been described as a key regulator of Ewing’s sarcoma, but the role of FOXO3 is less clear [27]. Preliminary studies showed that FOXO3 may also be a key regulator of Ewing’s sarcoma as well as a prognostic marker of uterine sarcoma [28,29]. In sum, results show that machine learning-enabled discovery is a viable and important avenue for finding key network genes regulating gene dependencies. 

## 3. Discussion

Gene dependencies allow for the systematic quantification of gene importance in a cellular context. Practically this guides the development of novel therapeutics and understanding of disease growth phenotypes [1]. While measuring single dependencies is feasible for a limited number of cell lines, high-dimensionality screens on numerous tumors or tumor types require extensive time and resources. Further, while in vitro gene dependency measurement reveals information about the measured gene, it does not reveal the macroscopic regulators of growth or mechanisms that control or determine gene dependency. Our ability to use gene dependencies to infer cellular behaviors and identify which genes and gene-gene interactions are key to disrupting a disease phenotype can help guide novel therapeutics. Here, through interpretable machine learning methodologies, we demonstrate first that network features add highly predictive information to dependency prediction models and second that this pipeline can reveal regulatory mechanisms present in specific tumor histologies. 

Both our pan-cancer and specific cancer subtype models were robust in predicting binary gene dependency. Models trained on network features had comparable predictive power to models trained on gene expression, indicating that these features can accurately summarize the systemic impact of each gene dependency. By integrating how a dependency relates to different cancer genes, hallmarks, and intermediates in the regulatory architecture, network features effectively add more layers of mechanistic information on top of what gene expression can offer. The major detractor of model accuracy was the diversity of cancers within the subtype. This underlines the importance of proper tumor stratification into meaningful prognostic and biological groups, which is also now being aided by machine learning [30,31,32]. The approach described here is broadly generalizable and thus can be adapted to changing cancer classifications or identifying sub-signatures related to specific functions within gene dependency. 

Our results also reveal interesting findings related to network topology and challenge the value of advanced centrality measures. High centrality is seen with many known cancer drivers, but advanced centrality measures were not found to be significant predictors of dependency in any cancer [33]. Indeed, dependent genes are thus often determined by the context of surrounding disease genes and not by being disease genes themselves [34,35,36]. Instead, our analysis points to the importance of the degree of connectivity and the sum of those connectivities’ strength over centrality measures beyond the local region. A more broadly increasing amount of attention is on the subnetwork’s ability to represent patient outcomes and disease biology [31,37,38]. Given our results, we hypothesize that gene dependency may be a subnetwork-level phenomenon as well, at least in some tumor types. 

Finally, we integrated publicly available cancer data with network topology to create novel biologically informed features. Numerous databases provide a broad overview of cancer biology, and network topology allows for specific measurements of genes within unique networks [17,18,39]. Here, we combined them to get the best of both worlds, a gene-level disease-specific measurement of well-known cancer phenomena. The resultant highlighted features show which broad cancer properties are at play in specific tumor types’ regulation of gene dependency. Several of these recovered markers have been described as determinants of disease progression or prognosis, but none have been examined for their role in determining gene dependency. Further, gene dependency coordination as a whole is understudied, but our results indicate that dependency coordination and its regulation is tumor-type specific. 

As network features illustrate the topological relationship between dependencies, markers, and intermediate genes between them, they can enable the testing of specific mechanistic hypotheses on how cancer gene dependency relates to essential cancer processes. With modern feature importance metrics such as SHAP SHapley Additive exPlanations scoring, it is possible to deploy interpretability measures in almost any AI model and identify which features most contributed to predicting each individual gene dependency in a cancer lineage. Such analyses could enable an exploration of how cancer processes involve or create gene dependencies. The redundancy in predictive information between pure topological features (centrality) and biologically driven features (hallmark and smallest/shortest paths to cancer genes) suggest that underlying processes related to both drive cancer gene dependency. In a similar vein, resolving the redundancy between expression and network features can provide more fine-grained explanations of the emergence of cancer gene dependency [39].

In light of our results, we posit that network features can encode valuable and complex biological contexts by adding information about the interaction network. We thus anticipate that these features could increase the accuracy of more complex models in the future. Here we used binary gene dependency for rapid model testing and deconstruction and achieved high accuracies. However, continuous dependency prediction remains a persistent challenge that network biology-informed machine learning may be poised to help solve. Finally, we show that network topology alone and biologically informed network topology reveal interpretable features that can predict dependency. By analyzing these features, we reveal several tissue-specific regulatory network motifs, pathways, and genes that can inspire new avenues of combinatorial therapy or sensitizing agents. Given our findings, we anticipate that machine learning-aided pharmacology could immensely benefit from informative and interpretable network-encoded features that represent both network topology and known biological information.

## 4. Materials and Methods

### 4.1. Key Packages

Pandas version 2.0.1, NumPy version 5.3.0., NetworkX (https://networkx.org/documentation/networkx-1.9/reference/generated/networkx.MultiGraph.edges.html accessed on 1 January 2023), Arboreto (https://github.com/aertslab/arboreto/blob/master/arboreto/algo.py accessed on 1 January 2023), and Scikit-learn (https://scikit-learn.org/stable/install.html accessed on 1 January 2023) were used to complete this work in Python3 and BiocManager (https://cran.rstudio.com/web/packages/BiocManager/readme/README.html accessed on 1 January 2023), minet, and igraph in R.

### 4.2. Data Sources and Preprocessing

Gene dependency data were obtained from the Cancer Gene Dependency Map (DepMap 21Q3, https://depmap.org/portal/download/all/ accessed on 24 May 2021), and corresponding expression data were obtained from the Cancer Cell Line Encyclopedia (CCLE) [21,40]. Cell lines with recorded values for all transcripts present (*n* = 16,379 transcripts) in the dataset and were also indexed in the 1K cosmic cell lines were retained for further use (*n* = 408 cell lines) [41]. Cancer type annotations were retained from the CCLE, and cancer types that were populated with less than three cell lines were excluded, leaving 21 valid cancer types. 

### 4.3. Cancer Type Network Construction

Each cancer type had a network constructed from its expression profiles using the ARACNE algorithm [15]. Edges with weights under MI < 1.5 were removed to concentrate the networks down to high-importance interactions. For oesophageal cancers, this threshold was increased to 2.0 due to the network’s large size to allow for a more reasonable run time. This pruning parameter was within the range of optimal values we found during our testing (Figure 3B).

### 4.4. Training and Testing Data

The previously formed graphs created the basis for training and testing data. For each cancer type-gene dependency combination, network features were measured, and training labels based on DepMap scoring were generated. The features fell into four categories: 1. classical network features, 2. cancer hallmark neighbors, 3. smallest weight (smallest) paths to cancer drivers, and 4. shortest paths to cancer drivers. 

Classical network topology: These features are measurements that have been used traditionally in the field of network topology. For this category, we measured the degree of node connectivity, the average degree of node connectivity of first neighbors, the sum of adjacent edge weights, betweenness centrality, and closeness centrality. 

Cancer hallmark neighbors: For the next category, we quantified the number of first-degree neighbors that fell under each of the Catalog Of Somatic Mutations In Cancer’s (COSMIC) Cancer Gene Census (CGC) hallmark pathways. This was used to quantify which cancer properties lie within the gene’s local network. 

Shortest Paths to CGC genes: The shortest path from the gene in question to all CGC genes was recorded to quantify how involved the gene being accessed is with key cancer drivers. CGC genes are assigned tiers of evidence by COSMIC. 

Smallest Paths to CGC genes: The smallest path from the gene in question, as measured by the sum of the edge weights within the path, to all CGC genes was recorded to quantify how involved the gene being accessed is with key cancer drivers. 

### 4.5. Labeling Gene Dependency

The label data for testing were the average of a gene’s CRISPR scores across the cell lines within each cancer type. The average DepMap scores were then converted to binarized data. Scores below or equal to −1 were annotated as dependent within the cancer type, and above −1 scores were assigned the label non-dependent. A threshold of −1 was chosen as DepMap scores were scaled such that −1 is the median score of ubiquitously essential genes across all cell lines, according to Dempster et al. [7,20,21]. This gave us a label of which genes were highly growth-restricting in each cancer type. 

### 4.6. Machine Learning Modeling and Feature Importance

AdaBoost, Decision Trees, Gaussian Process, K Nearest Neighbors, Linear Support Vector Classifiers (LVC), Logistic Regression, Random Forest, Ridge, Stochastic Gradient Descent optimization with LVC, and Radial Basis Function Support Vector Classifiers were tested in sci-kit learn using gene expression, our engineered network features, or a combination of both as features to predict dependency label. For all models, default hyperparameters as specified by sklearn were used. Testing was completed with 5-fold cross-validation, and accuracy was measured by F1 score because of the class imbalance between dependent vs. non-dependent genes. Average and cancer-specific performance was recorded.

For future testing, we focused on logistic regression models due to their interpretability, speed, and performance, which were among the top performers. In order to ascertain the importance of each feature class, model training with one class removed was completed for each of the four feature classes, and the change in model accuracy was used as a measure of feature class importance. Following this, pairs of feature classes were removed to determine if interactions between specific feature classes were key to performance. Specific feature importance within the class was determined using permutation importance values as implemented by sci-kit learn.

### 4.7. Testing Hyperparameter Effects on Efficacy

To test the stability of network AI models with respect to varying hyperparameters, we created models with a variety of hyperparameter values. First, the importance of edge weight pruning, which impacts network size, was tested by adjusting the value for pruning the ARACNE networks. The MI score cutoffs tested were 0, 0.5, 1.5, 1, 2, 3, 4, 5, 10, 15, 20, and 25. Following this, the DepMap cutoff for assigning classification labels was also tested, with cutoff values varying from −1.5 to 0.1 in 0.1 increments. For these, accuracy was plotted against the hyperparameters to visualize stability. 

## Figures and Tables

**Figure 1 pharmaceuticals-16-00752-f001:**
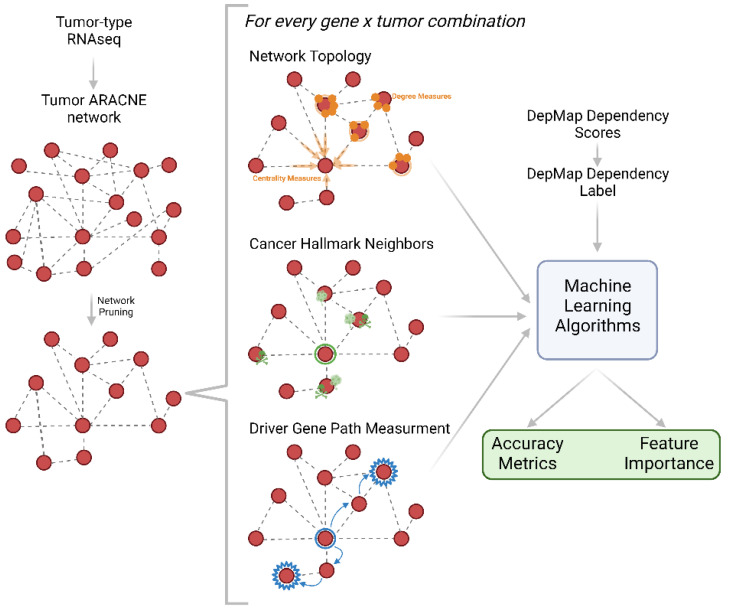
Schematic representation of analysis pipeline. RNAseq from each tumor type is used to create an ARACNE network which is then measured for network topology and network biology parameters. These parameters are used as inputs into machine learning algorithms which output accuracy metrics and feature importance.

**Figure 2 pharmaceuticals-16-00752-f002:**
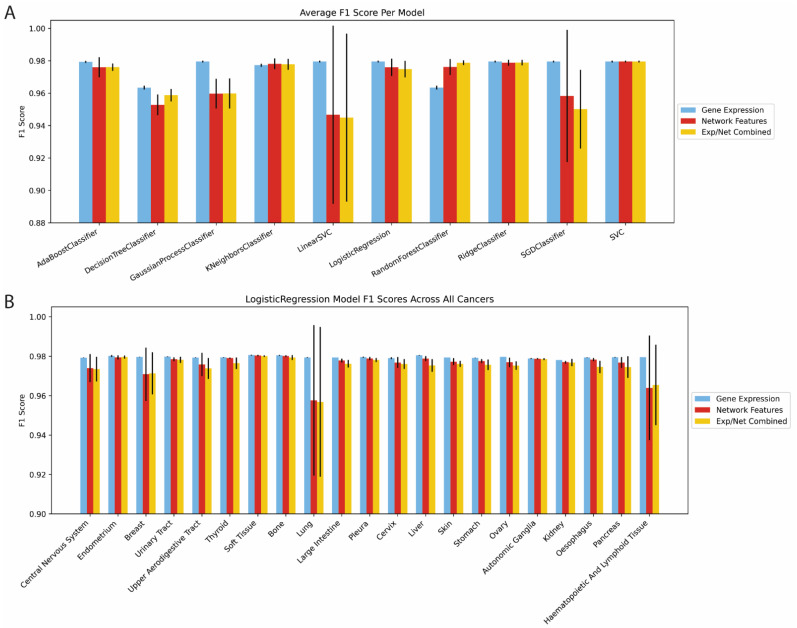
Baseline Accuracy Characterization. (**A**) F1 accuracy across different machine learning algorithms using gene expression, network features, and both combined. (**B**) F1 accuracy metrics in cancer-specific logistic regression models using gene expression, network features, and both combined.

**Figure 3 pharmaceuticals-16-00752-f003:**
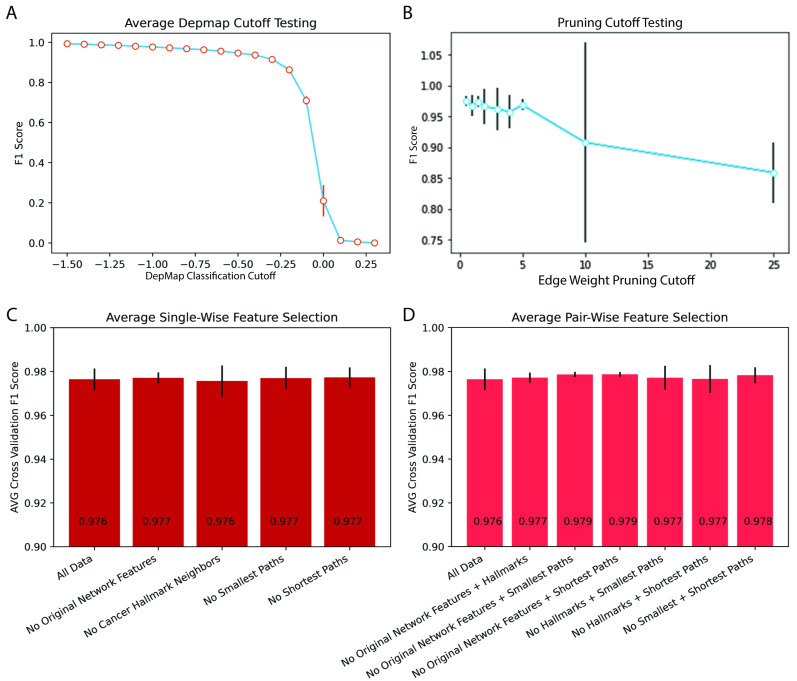
Hyperparameter Tuning and Ablation Models. (**A**) F1 accuracy versus DepMap cutoff determining dependent versus non-dependent genes. (**B**) F1 accuracy versus ARACNE edge weight cutoff for network pruning prior to network feature measurement. (**C**,**D**) F1 score versus feature ablation models with (**C**) single and double (**D**) gene ablation.

**Figure 4 pharmaceuticals-16-00752-f004:**
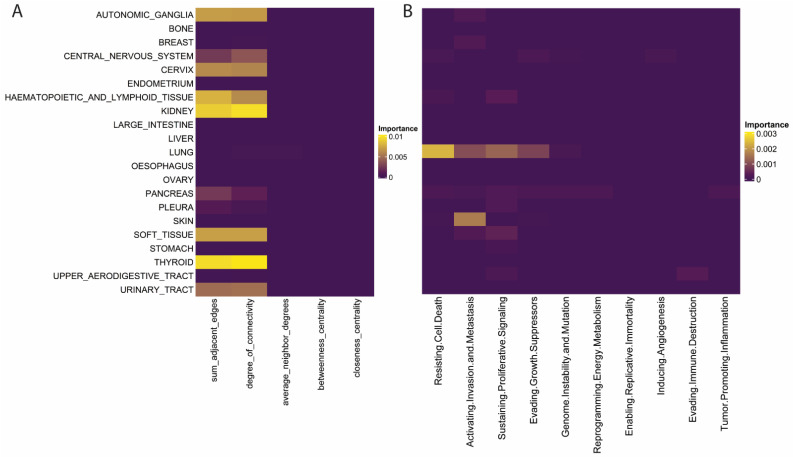
Feature Importance for Traditional Network and Cancer Hallmark Features. (**A**,**B**) Feature importance by tumor class for (**A**) traditional network and (**B**) cancer hallmark features.

**Figure 5 pharmaceuticals-16-00752-f005:**
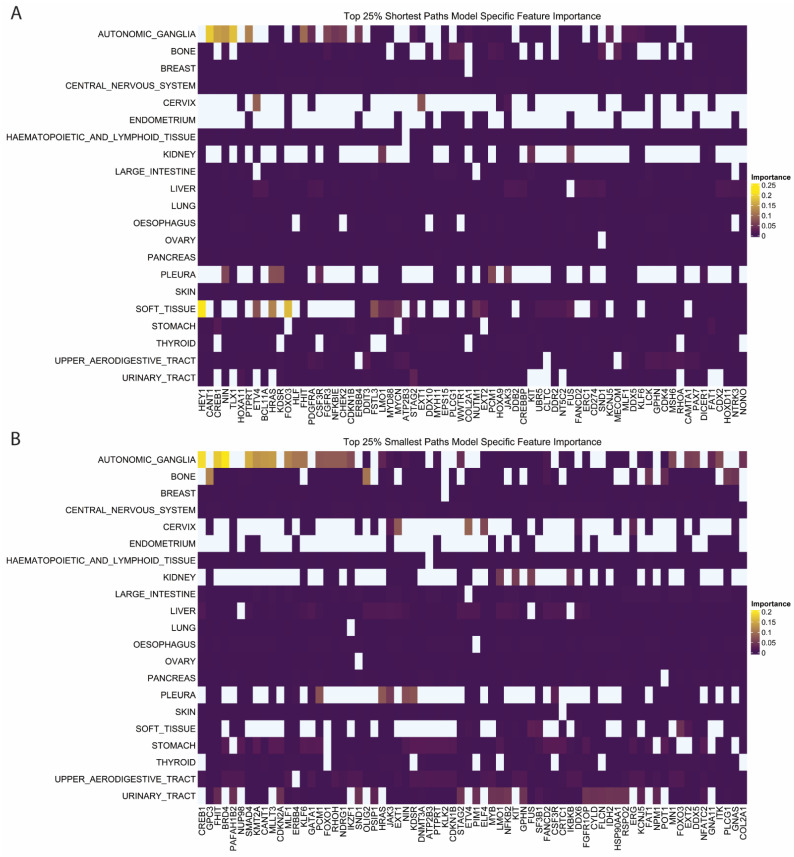
Feature Importance for Shortest and Smallest Path Features. (**A**,**B**) Feature importance by tumor class for (**A**) shortest path to cancer gene census genes and (**B**) smallest path to cancer gene census genes. The topmost important features are included. Extended plots shown in Appendix A.

## Data Availability

Data is contained within the article and Appendix A.

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
