# Peer review of "Network Biology-Inspired Machine Learning Features Predict Cancer Gene Targets and Reveal Target Coordinating Mechanisms"

_pharmaceuticals, 2023, doi:10.3390/ph16050752_

Round 1
Reviewer 1 Report
The authors have presented a well-written manuscript that combines machine learning and network biology to create robust algorithms for predicting gene dependencies in cancer, as well as identifying coordinating network features. This work provides valuable insights for cancer therapeutics development. However, there are concerns regarding the clarity of some figures, specifically Figure 1 and Figure 2, which need to be redone for better visibility. Addressing this issue would make the manuscript suitable for acceptance.
Author Response
The authors have presented a well-written manuscript that combines machine learning and network biology to create robust algorithms for predicting gene dependencies in cancer, as well as identifying coordinating network features. This work provides valuable insights for cancer therapeutics development. However, there are concerns regarding the clarity of some figures, specifically Figure 1 and Figure 2, which need to be redone for better visibility. Addressing this issue would make the manuscript suitable for acceptance.
We thank the reviewer for his/her thoughtful comments on our manuscript entitled “Network biology inspired machine learning features predict cancer gene targets and reveals target coordinating mechanisms” (Manuscript number: 2352850) by Weiskittel et al. We have replotted and incorporated high resolution Figures 1 and 2 in the revised manuscript. We believe this will address the reviewer’s concerns.
Reviewer 2 Report
This manuscript describes an interesting approach to predicting cancer gene dependency based on combining machine learning and network biology approaches. The approach moves beyond mere network topology or protein-protein interactions and incorporates connections to known cancer genes. Although the approach has promise, the manuscript is unclear in places and should address the following concerns.
A dependency score cut-off of -1 was used, despite the fact that it did not produce a better F1 than a cut-off of -0.25. However, it could have a different precision or recall. Why was -1 used?
No details are given as to how the cut-off was found. For example, was the cut-off found in a training dataset and evaluated in a testing dataset? How were these datasets created?
Lines 153-155 state “The lack of performance drop at smaller cutoffs showed that logistic regression was able to effectively ignore irrelevant edges without pruning”. It’s unclear what this is referring to, because in the logistic regression only network statistics are used, rather than the edges from a network.
Lines 205-207 state “Here we see that these vulnerabilities are likely determined by their proximity to the identified cancer hallmarks which are broadly coordinating gene dependencies in lung cancer.” This statement seems overly strong based on the evidence.
On lines 220-222, the statement “CREB1 has not been specifically interrogated as a drug target in these cancers, but our analysis, by recovering it from two separate feature sets, corroborates existing research that it plays a crucial regulatory role in autonomic tumors,” needs a citation.
On lines 252-254, the manuscript says, “This underlines the importance of proper tumor classification which is also now being aided by machine learning so that more accurate prognostic and therapeutic groups can be established,” However, this has been being done for some time.
Given that the network features did not add to gene expression features, the manuscript should do a better job of explaining the value of network features above what can be gotten from expression features.
The mutual information threshold used was arbitrary and changed for esophageal cancers. If a specific network size is recommended, then the threshold should slide to find that size network.
Stochastic Gradient Descent is listed as a machine learning method, but it is only an optimization method. Which machine learning method was optimized by stochastic gradient descent?
How were hyperparameters determined? No mention of a nested cross-validation is given.
The range of DepMap cutoffs tried was stated differently in Methods than earlier in the paper (lines 133 and 356).
Overall the quality is good, but there are appear to be editing mistakes in spots throughout the paper. Please give it another read through.
Author Response
We thank the reviewer for the very insightful comments on our manuscript entitled “Network biology inspired machine learning features predict cancer gene targets and reveals target coordinating mechanisms” (Manuscript number: 2352850) by Weiskittel et al. We have addressed each individual comment and revised the manuscript accordingly. Overall we believe our manuscript significantly improved upon revision.
This manuscript describes an interesting approach to predicting cancer gene dependency based on combining machine learning and network biology approaches. The approach moves beyond mere network topology or protein-protein interactions and incorporates connections to known cancer genes. Although the approach has promise, the manuscript is unclear in places and should address the following concerns.
A dependency score cut-off of -1 was used, despite the fact that it did not produce a better F1 than a cut-off of -0.25. However, it could have a different precision or recall. Why was -1 used?
No details are given as to how the cut-off was found. For example, was the cut-off found in a training dataset and evaluated in a testing dataset? How were these datasets created?
We appreciate the chance to clarify this matter. Because there was not a change in accuracy on F1, we used -1 cutoff to be consistent with the DepMap study and the broader field community using the DepMap data. This is stated at line 109-111 and for clarity we reiterated this rational at line 143-145 and 360-366.
Lines 153-155 state “The lack of performance drop at smaller cutoffs showed that logistic regression was able to effectively ignore irrelevant edges without pruning”. It’s unclear what this is referring to, because in the logistic regression only network statistics are used, rather than the edges from a network.
Thank you for bringing up this point. We have clarified this paragraph which now states:
“Here we used a MI score of 1.5 for our initial testing and varied pruning cutoffs (0-25) which dictate the size of output networks. Predictive performance (F1) was not significantly affected by varying pruning from 0 to 25. Performance was highest and most consistent with pruning cut offs less than 5 (Figure 3B). The lack of performance drop at smaller cutoffs suggests that network features combined with logistic were robust to irrelevant edges in under pruned networks (Figure 3B).”
Lines 205-207 state “Here we see that these vulnerabilities are likely determined by their proximity to the identified cancer hallmarks which are broadly coordinating gene dependencies in lung cancer.” This statement seems overly strong based on the evidence.
We appreciate this feedback and have modified the language to be more realistic. It now reads as follows: “Here we see that these vulnerabilities are likely determined by their proximity to the identified cancer hallmarks, and genes within these hallmarks may be coordinating gene dependencies in lung cancers.”
On lines 220-222, the statement “CREB1 has not been specifically interrogated as a drug target in these cancers, but our analysis, by recovering it from two separate feature sets, corroborates existing research that it plays a crucial regulatory role in autonomic tumors,” needs a citation.
Apologies the quoted line was referring to citation 25 which is cited in the line above. The citation has been added to this sentence as well for clarity.
On lines 252-254, the manuscript says, “This underlines the importance of proper tumor classification which is also now being aided by machine learning so that more accurate prognostic and therapeutic groups can be established,” However, this has been being done for some time.
Thank you for this comment. We did not mean to use classification in this sense, and have expanded the description to be more precise:
“This underlines the importance of proper tumor stratification into meaningful prognostic and biological groups which is also now being aided by machine learning so that more accurate prognostic and therapeutic groups can be established.(29–31)”
Given that the network features did not add to gene expression features, the manuscript should do a better job of explaining the value of network features above what can be gotten from expression features.
We appreciate the request to clarify the impact of this work. We have elaborated on the importance of network features at new points within the results and discussion at lines 118-119, 173-176, 260-266, and 294-306.
The mutual information threshold used was arbitrary and changed for esophageal cancers. If a specific network size is recommended, then the threshold should slide to find that size network.
Thank you for this comment. For esophageal cancer, the cutoff was scaled back to improve run time as the models were shown to be robust to pruning parameters at both 1.5 and 2. This has been clarified in the methods.
Stochastic Gradient Descent is listed as a machine learning method, but it is only an optimization method. Which machine learning method was optimized by stochastic gradient descent?
We have revised this phrase to “Stochastic Gradient Descent optimization with LVC”
How were hyperparameters determined? No mention of a nested cross-validation is given.
Thank you for this comment. We now include the following section in the methods:
“For all models, default hyperparameters as specified by sklearn were used.”
Nested cross validation was not used in this work.
The range of DepMap cutoffs tried was stated differently in Methods than earlier in the paper (lines 133 and 356)
We apologize for the discrepancy and we have rectified this.
Reviewer 3 Report
The article titled" Network biology inspired machine learning features predict cancer gene targets and reveals target coordinating mechanisms", was a well written study with appropriate methodology, clear results and discussion. The authors have investigated a large-scale screens and their cancer gene connections. A Novel treatment development requires anticipating and understanding tumors' gene activity needs. They used DepMap to show that machine learning and network biology can reliably predict a cancer's genes and network properties. Also they created four novel engineered machine learning features using network biology to predict binary gene dependencies with good accuracy. The authors concluded that network features with a biological basis can be a powerful complement to predictive pharmacology models, illuminating the underlying mechanisms of drug action.
Overall, the article is of readers' interest and I suggest the acceptance of the paper; however, there are aspects in the article that needs to be adjusted:
-All figures quality need to be revised.
- What are the shortcomings of the methods they used with their findings. I suggest what coul be the future use or directions based on this study can be suggeted.
Best Regards,
Author Response
The article titled" Network biology inspired machine learning features predict cancer gene targets and reveals target coordinating mechanisms", was a well written study with appropriate methodology, clear results and discussion. The authors have investigated a large-scale screens and their cancer gene connections. A Novel treatment development requires anticipating and understanding tumors' gene activity needs. They used DepMap to show that machine learning and network biology can reliably predict a cancer's genes and network properties. Also they created four novel engineered machine learning features using network biology to predict binary gene dependencies with good accuracy. The authors concluded that network features with a biological basis can be a powerful complement to predictive pharmacology models, illuminating the underlying mechanisms of drug action.
Overall, the article is of readers' interest and I suggest the acceptance of the paper; however, there are aspects in the article that needs to be adjusted:
-All figures quality need to be revised.
- What are the shortcomings of the methods they used with their findings. I suggest what could be the future use or directions based on this study can be suggested.
Best Regards
We thank the reviewer for the very helpful comments on our manuscript entitled “Network biology inspired machine learning features predict cancer gene targets and reveals target coordinating mechanisms” (Manuscript number: 2352850) by Weiskittel et al. We have revised every single figure now included in the revised manuscript. Additionally, we have included a paragraph in the Discussion (lines 294-306) to provide insights on future directions.
Round 2
Reviewer 2 Report
The manuscript is greatly improved. However, with the clarifications, I have concerns about the methodology. Initially, a number of machine learning methods are compared and logistic regression is chosen because it performed well over all. However, all the other methods listed are sensitive to the choice of hyperparameters and only the default hyperparameters were used. Those methods require the choice of appropriate hyperparameters. As it stands, I'm unsure why the comparison was included if only default hyperparameters could be used. Indeed, it appears as if logistic regression works well enough to provide meaningful results. However, given the comparison among methods is included in the manuscript, it also seems as if that comparison should be appropriately done.
Author Response
Thank you for following up on your review of our manuscript.
The manuscript is greatly improved. However, with the clarifications, I have concerns about the methodology. Initially, a number of machine learning methods are compared and logistic regression is chosen because it performed well over all.
A minor point to clarify here, we chose logistic regression because it performed as well as the other methods, trains quickly, and can be easily interpreted. No one model was definitively superior.
However, all the other methods listed are sensitive to the choice of hyperparameters and only the default hyperparameters were used. Those methods require the choice of appropriate hyperparameters. As it stands, I'm unsure why the comparison was included if only default hyperparameters could be used. Indeed, it appears as if logistic regression works well enough to provide meaningful results. However, given the comparison among methods is included in the manuscript, it also seems as if that comparison should be appropriately done.
We appreciate this thoughtful comment and would like to provide two points that describe our thinking on these issues. Firstly, in regard to the comparison’s inclusion, we use all models not to show that one is superior over the other, but that network features, like raw omics features, are flexible enough to be used by many architectures. We have stated this explicitly in the revised text to clarify. In the gene dependency literature, numerous model types are used making model compatibility important. Second in regard to tuning, when looking at the accuracies, we would have tuned the hyperparameters, but as you can see in Figure 1 we did not have accuracy gains to tune towards as almost all models were highly accurate. Thus, we focused on understanding the novel features which we see as the major focus of the article. We anticipate using these features for more difficult and/or more deeply interpretable dependency related tasks such as multiclass (Ex: Highly Sensitive, Moderately Sensitive, Resistant, Very Resistant) or continuous dependency prediction. This will undoubtedly require more dedicated model building, but we wanted to first establish if network-based features are a useful tool in this effort.
Round 3
Reviewer 2 Report
The authors' argument for not performing hyperparameter tuning is acceptable in this case.